# Seismic Performance and Nonlinear Strain Analysis of Mechanical Splices for High-Strength Reinforcement in Concrete Structures

**DOI:** 10.3390/ma16124444

**Published:** 2023-06-17

**Authors:** Hung-Jen Lee, Tzu-Yu Chang, Chien-Chung Chen, Ker-Chun Lin

**Affiliations:** 1Department of Civil and Construction Engineering, National Yunlin University of Science and Technology in Taiwan, Yunlin 640301, Taiwan; 2E. S. Witchger School of Engineering, Marian University, Indianapolis, IN 46222, USA; 3National Center for Research on Earthquake Engineering, Taipei 106219, Taiwan

**Keywords:** mechanical bar splices, couplers, grouted splice sleeve, seismic performance, displacement ductility, acceptance test

## Abstract

This study investigates the strain development in reinforcing bars within the plastic hinge regions of beams and columns, with the main objective of modifying the current acceptance criteria for mechanical bar splices to accommodate high-strength reinforcement. The investigation utilizes numerical analysis based on moment–curvature and deformation analysis of typical beam and column sections in a special moment frame. The results indicate that the use of higher grade reinforcement, such as Grade 550 or 690, results in lower strain demands in the plastic hinge regions compared to Grade 420 reinforcement. To validate the modified seismic loading protocol, over 100 samples of mechanical coupling systems were tested in Taiwan. The test results demonstrate that the majority of these systems can successfully complete the modified seismic loading protocol and are suitable for use in critical plastic hinge regions of special moment frames. However, caution is advised for slender mortar-grouted coupling sleeves, as they were unable to fulfill the seismic loading protocols. These sleeves may be conditionally used in plastic hinge regions of precast columns, provided they meet specific conditions and demonstrate seismic performance through structural testing. The findings of this study offer valuable insight into the design and application of mechanical splices in high-strength reinforcement scenarios.

## 1. Introduction

The use of mechanical splices in place of lap splices for reinforcing steel bars in concrete structures has become widespread. This shift can be attributed to the capabilities of mechanical couplers or coupling sleeves, which facilitate the connections of prefabricated steel reinforcement cages or precast concrete elements on site, speeding up the construction of bridges and buildings. Over the past decades, reinforcement industries have shifted towards producing higher grade reinforcing steel bars, accompanied by the development of various types of mechanical couplers or coupling sleeves [1] as alternatives to lap and welded splices. Several international standards [2,3,4,5] provide corresponding test protocols and acceptance criteria for mechanical splices of regular strength reinforcement. Today, a number of reinforcement producers in Asia-Pacific regions are capable of producing high-strength Grade 550 and 690 reinforcements, where the number refers to the minimum specified yield strength ($f_{y}$) in MPa. In order to alleviate steel congestion in reinforced concrete structures located in high seismic regions, such as Japan and Taiwan, construction industries are exploring the utilization of higher grade reinforcements. Consequently, in accordance with the amendments of ACI 318-19 Code [6], the Ministry of the Interior of Taiwan is currently revising the upcoming Design Code of Concrete Structures [7] to permit the use of Grade 550 and 690 bars instead of Grade 420 bars for flexural reinforcement in special seismic frame and wall systems. These special seismic systems are anticipated to experience inelastic deformation reversals during a violent earthquake, resulting in excessive tensile stresses beyond the bar $f_{y}$ within plastic hinge regions. Therefore, positioning mechanical splices in plastic hinge regions is prohibited in most design codes [1], unless the mechanical splices have sufficient ductility to withstand stresses beyond $f_{y}$ and endure inelastic strains exceeding the nominal bar yield strain ($\varepsilon_{y}$).

The special provisions for earthquake-resistant structures in the upcoming Design Code of Concrete Structures [7] in Taiwan has been revised to align with the ACI 318-19 Code [6] in the United States, with an exception for the categories and requirements of mechanical bar splices. According to ACI 318 Section 18.2.7, mechanical splices are categorized as Type 1 for basic applications and Type 2 for earthquake resistance. Type 2 mechanical splices shall be capable of developing 1.25$f_{y}$ and the specified tensile strength ($f_{u}$) of the spliced bars. This requirement of 1.25$f_{y}$ and $f_{u}$ is intended to ensure a certain level of ductility and prevent splice failure when the spliced bars undergo yielding. However, for some demanding applications such as splices located within plastic hinges subjected to multiple inelastic deformation reversals, the strength requirement of 1.25$f_{y}$ and $f_{u}$ may be insufficient. Therefore, in accordance with Section 18.2.7.2 of ACI 318-19 Code [6], mechanical splices shall not be located within a distance equal to twice the member depth (*h*) from critical sections, which includes the faces of columns or beams in special moment frames or locations where yielding of the reinforcement is likely to occur. However, an exception allows for Type 2 mechanical splices on Grade 420 reinforcement to be permitted at any location. In other words, this provision requires that mechanical splices on Grade 550 and 690 reinforcements be placed at a distance of 2*h* away from the faces of columns or beams in special moment frames. As a result, the use of mechanical splices on Grade 550 or 690 reinforcements is not feasible in a typical column with a height-to-depth ratio of four or less in a special moment frame. To enable the use of mechanical splices in potential yield regions, Section 7.3.1.4 of the Applied Technology Council report [8] recommends further studies to establish strain-based instead of stress-based requirements for new Type 3 mechanical splices to ensure adequate bar elongation outside the connection.

Although the high-strength reinforcement for seismic applications has a ductile stress–strain curve [8], components reinforced with such high-strength reinforcement may still exhibit reduced ductility due to low-cycle fatigue [9,10]. Some special considerations of manufacturing and detailing are necessary due to the enhanced ductility of the structural components under seismic excitations. Hassan and Elmorsy [11] conducted a comprehensive review of previous cyclic tests conducted on a range of high-strength reinforced concrete components, including columns, beams, beam-column joints, walls, and coupling beams. A comprehensive database was established and evaluated. The researchers concluded that substituting lower grade steel with high-strength steel for longitudinal reinforcement, based on matching flexural strength in beams and columns, resulted in reduced hysteretic energy dissipation and drift capacity. Using high-strength steel as transverse reinforcement could achieve higher confining stress to enhance the strength and ductility in structural components. In addition to increasing confinement, Choi et al. [12], as well as Almasabha and Chao [13], demonstrated innovative configurations of reinforcement for achieving high-ductility coupling beams and squat walls.

Cyclic loading tests of reinforced concrete components utilizing mechanical bar splices can be found in the literature [14,15,16,17,18,19,20,21,22,23,24,25,26,27,28]. Notably, the majority of these component tests involved mechanical splices with bar Grade 420 or lower, except for the columns tested by Ou et al. [16] and Wardi et al. [19], which provided the most recent available data on the utilization of Grade 690 mechanical splices. Prior studies have shown promising results regarding the use of mechanical splices in plastic hinges of beams or columns. However, two beam-column joints tested by Ingham and Bai [14] displayed very pinched hysteretic response attributed to excessive coupler slippage. Ingham and Bai [14] concluded that “in-air” tensile tests of mechanical bar splices are unable to accurately demonstrate the performance of couplers in plastic hinge regions. Ji et al. [28] also tested three exterior beam-column joints for precast construction. Unfortunately, one of the specimens exhibited premature failure due to the unexpected pullout of the mechanical couplers. This unfavorable pullout failure was attributed to the incomplete tightening of the couplers. In addition, Ji et al. [28] conducted tensile tests on three rebar coupler samples with varying degrees of tightness. However, none of the samples were able to develop the bar tensile strength.

To bridge the gap between tests of mechanical bar splices and column connections, modeling of mechanical coupling sleeves in bridge columns can be found in References [29,30]. Tazarv and Saiidi [31] also conducted a comprehensive review of uniaxial tensile tests of couplers and cyclic tests of columns with couplers in plastic hinge regions. They proposed minimum requirements and simplified design equations for precast concrete column connections. Bompa and Elghazouli [22,32,33] conducted another series of tests of mechanical bar splices and column connections, finding that slender couplers affect plastic hinge behavior by localizing curvatures and reducing rotational capacity. Columns with compact couplers demonstrate cyclic responses similar to specimens with continuous reinforcement. However, these studies did not explore the utilization of Grade 550 and 690 mechanical splices.

Based on the literature review, the use of mechanical splices in plastic hinge regions can be considered feasible under certain conditions, including proper design, installation, and inspection of the mechanical splices. Mechanical splices are typically evaluated through “in-air” uniaxial tests on spliced assemblies. Extensive research has been conducted on different types of mechanical bar splices, using various loading protocols [33,34,35,36,37,38,39,40,41,42,43,44,45,46]. Although the “in-air” uniaxial tests do not accurately represent the behavior of mechanical splices embedded “in concrete” members [22,32,33], they are commonly adopted for testing and acceptance in construction practice. Component tests may be required for special applications or when utilizing new materials, such as Grade 690 mechanical splices in plastic hinge regions.

Among the test data on mechanical bar splices [33,34,35,36,37,38,39,40,41,42,43,44,45,46], the majority focused on bar Grade 500 or smaller, with the exception of Lee et al. [38], who investigated bar Grade 550 and 690. The evaluation of mechanical bar splices involved various loading protocols, including monotonic [34,40,43,45], code-specified cyclic [33,35,39,41,42,44,46], or researcher-defined protocols [36,37,38]. The most commonly used cyclic loading protocols for mechanical bar splices, categorized for basic, moderate, and violent seismic applications, are specified in ISO 15835 [2,3]. The categories and loading protocols of mechanical bar splices in the Japan Standard [4] and Chinese Standard [47] are similar to those specified in ISO 15835. Based on a review of the aforementioned standards, the authors proposed amendments to the upcoming Design Code of Concrete Structures [7], introducing Type 1 (B), Type 2 (A), and Type 3 (SA) categories. Table 1 provides a comparison of the categories, while Table 2 and Figure 1 illustrate the loading protocols. Since Taiwan is located in a highly seismic region, the majority of the practical applications fall under Category S2 or SA. While the categories of basic, moderate, and violent seismic applications are similar across standards [2,3,4,7,47], there are differences in the testing set-ups and loading protocols.

Category S1 of ISO 15835 (or Category A of JSCE 2007) aims to simulate the elastic cyclic loading experienced by a concrete structure under moderate earthquakes or wind loads. The test sample is subjected to 20 cycles, starting from zero stress to 0.90$f_{y}$ (or 0.95$f_{y}$ of JSCE 2007) in tension and then reversal to 0.5$f_{y}$ in compression, as illustrated in Figure 1. Category S2 (or Category SA) aims to simulate the inelastic cyclic loading experienced by a concrete structure under a violent earthquake. Category S2 (or Category SA) involves four cycles for each of the two post-yield strain amplitudes (2$\varepsilon_{y}$ and 5$\varepsilon_{y}$) with a corresponding reversal to 0.5$f_{y}$ in compression. After completing the specified cycles, each test sample should be loaded in tension until failure to determine its tensile strength.

The Design Code of Concrete Structures in Taiwan has incorporated a modified Category SA. It involves eight cycles for each of the two post-yield strain amplitudes (n$\varepsilon_{y}$ and 2n$\varepsilon_{y}$) with a corresponding reversal to 0.5$f_{y}$ in compression. A constant ductility ratio of n=6 for Grade 420 bar splices has been adopted over the past two decades. This modification results in higher ductility levels of 6$\varepsilon_{y}$ and 12$\varepsilon_{y}$, obviously exceeding the 2$\varepsilon_{y}$
and 5$\varepsilon_{y}$ requirements specified by ISO or JSCE. In Japan, the 2$\varepsilon_{y}$ and 5$\varepsilon_{y}$ requirements are extended to the use of Grade 690 mechanical splices. In Taiwan, there is ongoing debate regarding whether to extend the constant ductility ratio of n=6 for Grade 550 and 690 mechanical splices. Therefore, a thorough review of the aforementioned testing protocols and acceptance criteria for strength, ductility, and slip were completed by Lee et al. [38] and Chang [48]. The research objective of this paper was to estimate rational ductility ratios for the inelastic cyclic loading protocols of Category SA, particularly for the use of Grade 550 and 690 mechanical splices. A moment–curvature and deformation analysis approach is presented to estimate the expected strain demands on reinforcing bars in plastic hinge regions. Based on the analysis results, the ductility ratio of n is modified for Grade 550 and 690 mechanical splices. The test results of the mechanical bar splices using modified loading protocols of Category SA in Taiwan are summarized.

## 2. Analysis of Strain Demand for Reinforcing Bars in Plastic Hinge Regions

This study used the following steps to estimate the strain demands for reinforcing bars in the plastic hinge regions of beams and columns and proposed modifications to the loading protocols of Category SA in Taiwan.

Establish an analytical model to predict the behavior of concrete members under lateral cyclic loading and estimate the strain development in plastic hinge regions of longitudinal bars.Validate and calibrate the analytical model using selected components from prior experiments, incorporating measured material properties and strain readings.Apply the model to simulate virtual beam and column sections and analyze the trends of the strain demand of the longitudinal bars in the plastic hinge regions.Propose and validate the loading protocol for Category SA mechanical splices by testing samples of Grade 550 and 690 mechanical bar splices.

### 2.1. Moment–Curvature Analysis and Deformation Prediction

The load–deflection behavior of moment frame beams and columns can be analyzed using conventional flexural theory. This involves estimating and simplifying the curvature distribution along the member and integrating the curvatures to calculate rotations and displacements. The moment–curvature analysis is a widely accepted method for predicting the curvatures of reinforced members. To facilitate this analysis, Yu et al. [49] developed a new moment–curvature analysis program called NewRC-Mocur2020. The NewRC-Mocur2020, similar to Response 2000 [50] or XTRACT [51], provides a user-friendly interface for determining the moment–curvature relationship of a reinforced concrete cross-section subjected to moment and axial load. NewRC-Mocur2020 offers flexibility for users to define stress–strain models matching the measured material results. In addition to high-strength concrete and reinforcement models, one notable feature is the ability to set the critical stress for reinforcing bar buckling. The moment–curvature relationships can be derived by specifying the material models for reinforcing bars, cover (i.e., unconfined) concrete, and core (i.e., confined) concrete. The material models proposed by Lee et al. [52] and Mander et al. [53] were used in this study. Furthermore, the proposed model was validated by analyzing the beams tested by To and Moehle [54]. These tested beams are representative of beams used in special moment frames in accordance with ACI 318 code [6], except for the use of Grade 690 reinforcement instead of Grade 420. The stress–strain relationships of the materials and load–deflection curves of the tested beams can be found in Reference [54]. By comparing the results of the analysis with the measured data, the accuracy and validity of the modeling approach can be evaluated.

Figure 2 illustrates the plastic hinge model for a cantilever beam unit, where the length of the cantilever ($L_{s}$) represents the shear span of a frame beam under lateral sway. The lateral drift is generally attributed to three components: flexure, shear, and slip deformation. The flexure deformation can be determined using the second moment–area theorem with the idealized curvature distribution, as shown in Figure 2. The lateral drift at the beam tip beyond the yield to the ultimate state can be expressed as shown below.
(1)$$\Delta =\Delta_{y}+\Delta_{p}=\left(\frac{\phi_{y}L_{s}^{2}}{3}+\Delta_{v,y}+\Delta_{s,y}\right)+\left(\phi_{u}-\phi_{y}\right)\ell_{p}\left(L_{s}-0.5\ell_{p}\right)$$
(2)$$\phi_{y}=\frac{\varepsilon_{y}}{\left(d-c_{y}\right)}$$
(3)$$\phi_{u}=\frac{\varepsilon_{s}}{\left(d-c\right)}$$
where $\phi_{y}$ and $\phi_{u}$ are the yielding and ultimate curvature, respectively; d is the effective depth, while $c_{y}$ is the depth of neutral axis at yielding. The neutral axis c beyond the yield to the ultimate state is typically smaller than $c_{y}$ when the bar strain ($\varepsilon_{s})$ exceeds $\varepsilon_{y}$. Traditionally, an equivalent lump plasticity hinge length $\ell_{p}$ is assumed to account for the plastic rotation, including curvature, shear deformation, and slip [55,56,57,58]. For a special moment frame beam or column, $\ell_{p}=0.5$*h* is assumed to be simple and reasonably accurate [59] (p. 207) in the following analysis. More delicate expressions of $\ell_{p}$ can be found in References [57,59].

The shear component ($\Delta_{v,y}$) at yielding can be estimated by assuming the member with a constant shear modulus.
(4)$$\Delta_{v,y}=\frac{V_{y}L_{s}}{A_{v}G_{eff}}$$
where $V_{y}$ is the force corresponding to the yielding moment; $A_{v}$ is the effective shear area of five-sixths of $A_{g}$ for a rectangular section with a gross area of $A_{g}=bh$; $G_{eff}=0.2E_{c}$ is the effective shear modulus of the cracked concrete, as recommended by Elwood and Eberhard [60]; and $E_{c}=4700\sqrt{f_{c}^{\prime }}$ is the elastic modulus of the concrete, as given in ACI 318. For a flexure-dominated beam or column, the contribution of the shear deformation is relatively minor to the flexure component. It can be ignored to simplify the calculation.

The slip component ($\Delta_{s}$) at yielding is relatively significant and should be considered in deformation estimation [59] (p. 200).
(5)$$\Delta_{s,y}=\theta_{slip}L_{s}=\frac{slip}{\left(d-c_{y}\right)}L_{s}=\frac{f_{y}^{2}d_{b}}{8E_{s}u_{b}}\frac{L_{s}}{\left(d-c_{y}\right)}=\frac{d_{b}f_{y}}{8u_{b}}\phi_{y}L_{s}$$
where the slip of the bar (slip) with a diameter of $d_{b}$ can be determined from integrating the strains over the development length with an average bond stress of $u_{b}$. Several researchers [59] (p. 200) recommended values of $u_{b}$ ranging from $0.5\sqrt{f_{c}^{\prime }}$ to $1.0\sqrt{f_{c}^{\prime }}$ MPa. This study adopted $u_{b}=1.0\sqrt{f_{c}^{\prime }}$, as recommended by Sezen and Setler [56].

Bompa and Elghazouli [22] conducted a comprehensive experimental investigation on the inelastic cyclic performance of reinforced concrete members incorporating threaded reinforcement couplers. They utilized the plastic hinge model and sectional moment–curvature analysis to study the distribution of plasticity in tested columns, which were reinforced with continuous Grade 500 reinforcement or mechanically spliced reinforcing bars. In their analysis, the expressions for the plastic hinge length yielded values for $\ell_{p}$ of approximately 0.85 h, which were longer than the 0.5 h used in this paper. The predicted yield and ultimate rotations agreed well with the test data presented by Bompa and Elghazouli [22], with the exception of the tested column incorporating slender couplers. In that column, the coupler concrete slip altered the plastic hinge behavior and reduced the ductility.

In this paper, the bar strain demand was back calculated and compared with the measured data to demonstrate the validity of the plastic hinge model. The observations from this paper show that the proposed model can be used for estimating the strain demand in plastic regions.

### 2.2. Experimental Verification

Using Equations (1)–(5) with $\phi_{y}$ and $\phi_{u}$ determined by the moment–curvature analysis, the lateral load versus the lateral deformation at yield and beyond yield to the ultimate sate can be predicted. Figure 3 compares the lateral load–deformation predictions to the four beams tested by To and Moehle [54]. In the test program, the reference Specimen SBH60 used Grade 60 ksi (420 MPa) as the primary reinforcement, with a tensile-to-yield strength ratio (T/Y) of 1.48 and uniform elongation ($\varepsilon_{su}$, strain at maximum stress) of 0.114. The Specimens SBH100 and SBL100 used a Grade 100 ksi (690 MPa) reinforcement, which had reduced a T/Y of 1.26 and 1.17, respectively, and a reduced $\varepsilon_{su}$ of 0.094 and 0.068, respectively. Unlike other specimens using a reinforcement with a well-defined yield plateau, Specimen SBM100 used an ASTM A1035 Grade 100 ksi (690 MPa) reinforcement with a roundhouse curve with T/Y = 1.38 and $\varepsilon_{su}$ = 0.056, which is less than the minimum $\varepsilon_{su}$ of 8% recommended in Reference [8]. For each specimen, the moment–curvature analysis was significantly affected by the stress–strain relationship of the reinforcement. By using the measured material properties into the moment–curvature analysis and plastic hinge model, the global load–deformation response can be accurately predicted, as demonstrated in Figure 3.

Using Equation (3), the local response of the bar strain ($\varepsilon_{s}$) at the beam–joint interface, which is the critical section of the maximum moment at the fixed end of the beam, can also be determined using back calculation from the curvature ($\phi_{u}$) and neural axis depth (c) at each step of the drift ratio. Figure 4 compares the measured and modeled strain development of the longitudinal bar at the critical beam–joint interface for the four beams. Notably, among the tested specimens, only Specimens SBH60 and SBH 100 used reinforcement with a well-defined yield plateau and a T/Y ratio greater than 1.25. It is important to satisfy the specified T/Y ratio of 1.25 to ensure an even distribution of the plasticity throughout the plastic hinge regions. Figure 4a illustrates that the strain value of approximately 2.5% (equivalent to 12$\varepsilon_{y}$ for bar $f_{y}$ = 420 MPa) was measured at a 3% drift ratio for Specimen SBH60. On the other hand, Figure 4b shows a slightly higher strain value of 2.6% (equivalent to 7.6$\varepsilon_{y}$ for bar $f_{y}$ = 690 MPa) at a 3% drift ratio for Specimen SBH100. It is suggested that using an equal strain value criterion rather than a constant ductility ratio of 6 or 12 may be more appropriate. According to the current code in Taiwan, a Category SA mechanical bar splice is required to complete an inelastic cyclic test of eight cycles up to a ductility of $2n\varepsilon_{y}$ = 12$\varepsilon_{y}$ for bar $f_{y}=420$ MPa. However, specifying the same ductility of 12$\varepsilon_{y}$ for a Grade 550 and 690 reinforcement may be excessively demanding.

Comparing the experimental and analytical results shown in Figure 3 and Figure 4, the model underestimated the yielding drift ratio ($\Delta_{y}$) but slightly overestimated the rate of strain development beyond yielding. This can be attributed to the fact that the moment–curvature analysis with the plastic hinge model is monotonic, whereas the tested beams experienced inelastic cyclic reversals. However, the proposed model is still conservative in estimating the strain demand at the critical section of special moment frame beams.

## 3. Ductility Demands for Reinforcing Bars in Plastic Hinge Regions

### 3.1. Simulations of Beam and Column Sections in a Special Moment Frame

Using the proposed model, it is possible to calculate the relationship of bar strain demands versus drift ratios at critical sections of reinforced concrete elements. In order to investigate the potential strain demand in a special moment frame, this study conducted simulations on a total of 72 beam sections and 81 column sections, as shown in Figure 5. The design parameters of the beam and column sections are listed in Table 3 and Table 4, respectively. The beam sections have a width of b= 600 mm and a section depth of h= 900 mm with code-conforming transverse reinforcement, as shown in Figure 5a. The beam simulations encompass 3 × 3 × 4 × 2 =72 combinations of the parameter options listed in Table 3. Similarly, as shown in Figure 5b and Table 4, the simulated column have a cross-section of 1000 × 1000 mm with code-conforming transverse reinforcement. The column simulations encompass 3 × 3 × 3 × 3=81 combinations of the parameter options. A beam span of 9 m and a column height of 3.6 m were selected for the design of a typical building frame. By assuming a double curvature deformation in beams or columns in a frame under sway, the shear span $L_{s}$ used in the plastic hinge model (Figure 3) can be taken as 9/2 = 4.5 m for the simulated beams and 3.6/2 = 1.8 m for the simulated columns. The selection of these parameters is informed by the experience of engineers in building structure design.

The moment–curvature response and bar strain demand of a beam or column section are strongly influenced by the stress–strain curves and T/Y ratio. As shown in Figure 5, Specimen SBL100 exhibited the highest rate of bar strain demand as the drift ratio increased, and this is attributed to the reduced T/Y = 1.17 for the reinforcement. In contrast, Specimen SBM100 displayed the lowest rate of bar strain demand at the critical section because it had the highest T/Y = 1.48. Therefore, in this study, a minimum T/Y ratio of 1.25 was selected for the simulations to ensure conservative results with higher bar strain demands.

Figure 6 shows the idealized stress–strain curves used in the simulations for Grade 420, 550, and 690 reinforcements with T/Y = 1.25. The curves were based on a Young’s modulus of 200 GPa and defined yield plateau corresponding to the yield strength $f_{y}$. Strain–hardening is assumed to initiate at a strain of 0.01 and reach the ultimate strength at $\varepsilon_{su}$ (uniform elongation), which was set as 0.12 for Grade 420, 0.11 for Grade 550, and 0.10 for Grade 690 bars. The ascending curves of the strain hardening used a third-order power function. The specific values for the strain–hardening curves were calibrated by fitting to them to tensile test samples of Grade 420, 550, and 690 reinforcements [48].

### 3.2. Strain Demands of the Simulated Sections in Plastic Hinge Regions

Figure 7 shows the relationships of the bar strain at the critical section versus the drift ratio for the simulated 72 beams and 81 columns. The curves in red, blue, and green correspond to the bar grades of 420, 550, and 690, respectively. Figure 7 demonstrates a clear trend, with the higher grade reinforcement exhibiting larger yield drift ratios followed by relatively shorter ultimate strain in the beams and columns. This implies that the strain demand for Grade 550 and 690 was slightly lower than that of Grade 420 reinforcement at limit state drift ratios, such as 2% or 3%. Considering the minimum T/Y = 1.25 used in this study, the rate (slope) of the strain development shown in Figure 7 is likely to be overestimated. In practical applications with Grade 420 reinforcement, the mean T/Y ratios are approximately 1.33 in the United States and 1.45 in Taiwan. Therefore, the actual strain demand would very likely be lower for the reinforcements with higher T/Y ratios.

### 3.3. Effects of Bar Yield Strength on Moment, Curvature, and Strain Demands 

Several cases of simulated beams and columns were selected to examine the effects of varying bar $f_{y}$ on moment–curvature behavior and strain demands. Figure 8 illustrates three beams that share the same concrete strength and reinforcement ratios but differ in bar $f_{y}$. As expected, a higher bar $f_{y}$ leads to increased moment and curvature at yielding, followed by a reduced ultimate curvature. Table 5 provides the bar strain and drift components at 3% drift for each beam. For the simulated frame beam with $L_{s}$ = 4.5 m, the lateral displacement Δ=0.03×4500 = 135 mm at 3% drift. Using proposed model and Equation (1), the lateral displacement Δ is attributed to yield deformation $\Delta_{y}=\Delta_{f,y}+\Delta_{v,y}+\Delta_{v,y}$ and plastic hinge contribution $\Delta_{p}$. Table 5 listed the shear, slip, and flexure components at yielding and $\Delta_{p}$ contributions to a 3% drift with respect to the bar strain ($\varepsilon_{s}$). It can be observed that as the bar $f_{y}$ increased, the $\varepsilon_{s}$ developed at 3% drift decreased due to the shorter $\Delta_{p}$ and longer $\Delta_{y}$. Additionally, increasing $f_{y}$ from 420 to 550 or 690 MPa not only amplified the curvature component proportionally but also increased the slip component, as indicated in Equation (5). The slip components $\Delta_{s,y}$ were more significant for the beams using higher grade reinforcement, while the shear components $\Delta_{v,y}$ were relatively minor in the proposed model.

In practical design, replacing Grade 420 reinforcement with Grade 690 or 550 reinforcement can reduce the amount of reinforcement for a given design force. By replacing Grade 420 #11 (D36) bars with Grade 550 #10 (D32) or Grade 690 #9 (D29) bars in a given section, the effect of changing bar $f_{y}$ can be compared for beams with equal moment strength. Figure 9 and Table 6 demonstrate the effects of changing bar $f_{y}$ and sizes for the reinforced beam sections. Notably, increasing bar $f_{y}$ and reducing bar sizes from 420 #11(D36) to 550 #10(D32) or 690 #9(D29) amplifies the curvature component proportionally. However, the slip component $\Delta_{s,y}$ is relatively smaller compared to previous results in Table 5, mainly due the effect of the bar diameter (Equation (5)). The $\varepsilon_{s}$ that developed at a 3% drift for all cases remained at approximately 3%. Once again, using a higher bar $f_{y}$ resulted in lower $\varepsilon_{s}$ due to the shorter $\Delta_{p}$ and longer $\Delta_{y}$ components.

Similarly, the use of Grade 690 or 550 reinforcement in place of Grade 420 reinforcement can reduce the amount of reinforcement in a column section resisting a combination of axial and bending moment. It can be understood that the higher axial load ratios would reduce the sectional curvature and result in a lower strain demand in reinforcing bars. Therefore, relatively low axial load ratios of 0.1, 0.2, and 0.3 are selected in this study. Figure 10 and Table 7 illustrate the column section reinforced with 32 #11(D36) ($f_{y}$ = 420 MPa), 32 #10(D32) ($f_{y}$ = 550 MPa), or 32 #9(D29) ($f_{y}$ = 690 MPa) bars. These sections have a reinforcement ratio of 3.22%, 2.61%, and 2.07%, respectively, while maintaining similar moment strength. The strain developments with the increase in the drift ratios are shown in Figure 10b. For the simulated frame column with $L_{s}$ = 1.8 m, the lateral displacement Δ=0.03×1800 = 54 mm at a 3% drift. Using the proposed model and Equation (1), the bar strain and drift components at a 3% drift for the column sections are listed in Table 7. The $\varepsilon_{s}$ that developed at a 3% drift was approximately 3.5% for all cases. Similar to the beams, increasing the bar $f_{y}$ resulted in lower $\varepsilon_{s}$ values due to the shorter $\Delta_{p}$ and longer $\Delta_{y}$ components. However, the effect of changing the bar $f_{y}$ on the columns was less significant compared to the beams, primarily due to the column axial force and shorter shear span.

In conclusion, increasing the bar $f_{y}$ did not lead to an increase in the bar strain ($\varepsilon_{s}$) developed at a target drift ratio. Instead, the strain demand was somewhat reduced when Grade 690 or 550 reinforcement was used instead of Grade 420 bars. This reduction is attributed to the larger slip and curvature deformation at a yielding associated with higher grade reinforcement.

## 4. Modified Loading Protocols for Mechanical Bar Splices under Seismic Conditions

Based on the presented numerical simulations, the bar strain ($\varepsilon_{s}$) that developed at a target drift ratio in the plastic hinges of the special moment frames were similar for the Grade 420, 550, and 690 reinforcements. This suggests that the constant strain ductility ratios of 12$\varepsilon_{y}$ currently used in Taiwanese practice may not be adequate for Grade 550 and 690 reinforcements. This study proposes an equal strain criterion instead of the constant ductility ratios of 12. This proposal has been adopted in the upcoming Design Code of Concrete Structures [7]. As shown in Table 2, the seismic loading protocol for Category SA mechanical splices has been modified to include two post-yield strain amplitudes of $n\varepsilon_{y}$ and $2n\varepsilon_{y}$ each for eight cycles with strain ductility ratios of n=6, 5, and 4 for Grade 420, 550, and 690 bar splices, respectively. This modification would result in an equal strain level of $2n\varepsilon_{y}$ = 2.75% for the inelastic cycles of Grade 550 and 690 bar splices.

The validity of the modification on the Category SA loading protocols was verified by sampling and testing 105 couplers or coupling sleeves of Grade 550 and 690 reinforcements, as shown in Table 8. Complete information on the test samples and results are reported in Reference [38]. Four types of mechanical coupling systems (Figure 11) were evaluated using the modified Category SA loading protocols. The test samples used high-strength and larger diameter bars, including #8 (D25), #10 (D32), #11 (D36), #12 (D38), and #13 bars (D41). The typical elastic and inelastic cyclic responses of the tested samples are shown in Figure 12. With the exception of some of the mortar-grouted coupling sleeves, which could not fracture the spliced bars due to the bar pull out (i.e., bond failure), all of the other samples were able to fracture the bar outside of the coupler region. The majority of the samples completed the elastic and inelastic cycles and satisfied the modified Category SA requirements. The test results demonstrated that the sampled couplers can be used to connect the primary reinforcement in the critical plastic hinge regions of the special moment frames. Most of the test samples exhibited minimal slip during the cyclic loading, except for the grouted couplers with thread-deformed bars. These couplers showed pinching hysteresis during the inelastic cycles (Figure 12c), indicating that some slip occurred. Additionally, some mortar-grouted coupling sleeve systems (Figure 12d) did not satisfy the Category SA requirements due to pull out failure during the inelastic cycles of $2n\varepsilon_{y}$. The failure was attributed to the slenderness of the coupling sleeve, which resulted in a long gauge length for the tensile test [34]. During the inelastic cycles, the inelastic strain concentrated at the spliced bars adjacent to the coupling sleeve, leading to yield penetration and bond damage along the spliced bars into the sleeve [61]. Therefore, the cyclic tests of the mechanical splice samples in air were conservative. Such mortar-grouted coupling sleeves may be used conditionally in plastic hinge regions of precast columns, subject to meeting certain conditions and demonstrating the seismic performance of columns through structural testing, as shown in References [15,16,17]. Special design considerations of such precast column connections can be found in References [30,31,36].

## 5. Conclusions

In conclusion, a numerical analysis of the strain development for reinforcing bars in plastic hinge regions of beams or columns, as well as an evaluation of the mechanical splices of high-strength reinforcement, provided advanced insight into the use of such systems in special moment frames. An evaluation of over 100 samples’ couplers and coupling sleeves using the proposed seismic testing protocols of Category SA led to the following key findings and recommendations:The use of higher grade reinforcement, such as Grade 550 or 690, can result in lower strain demands in plastic hinge regions compared to Grade 420 reinforcement because of the shorter plastic hinge length and larger slip and curvature deformation at yielding.This study recommends a modified Category SA loading protocol for the upcoming Design Code of Concrete Structures [7] in Taiwan, with two post-yield strain levels of two post-yield strain amplitudes of $n\varepsilon_{y}$ and $2n\varepsilon_{y}$ each for eight cycles. This protocol is very stringent for mechanical coupling systems available in Taiwan and Japan. The ductility ratio n should be adjusted based on the reinforcement grade.The test samples of the couplers and coupling sleeves demonstrated their suitability for satisfying modified Category SA loading protocols and should be permitted to be positioned in the critical plastic hinge regions of special moment frames.Mortar-grouted coupling sleeves should be used conditionally in plastic hinge regions of precast columns, subject to meeting specific conditions and demonstrating seismic performance through structural testing.These findings and recommendations provide valuable guidance for the design, testing, and use of mechanical splices in high-strength reinforcement applications, particularly for special moment frames in high-seismic zones.

## Figures and Tables

**Figure 1 materials-16-04444-f001:**
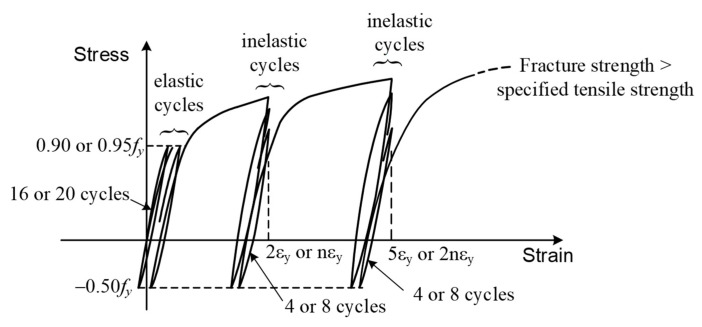
Illustrations of elastic and inelastic cyclic loading protocols for Table 2.

**Figure 2 materials-16-04444-f002:**
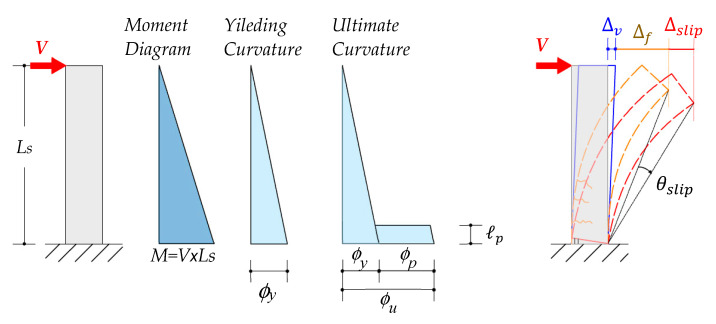
Plastic hinge model with the idealized curvature at the yield and ultimate states.

**Figure 3 materials-16-04444-f003:**
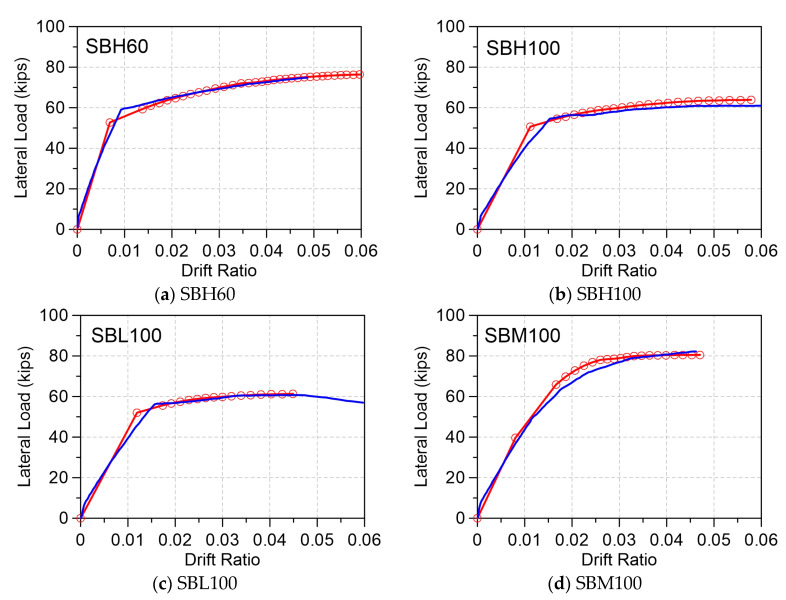
Plastic hinge model predictions (red) and the lateral load–deformation curves (blue) tested by To and Moehle [54]: (**a**) Specimen SBH60; (**b**) Specimen SBH100; (**c**) Specimen SBL 100; (**d**) Specimen SBM 100.

**Figure 4 materials-16-04444-f004:**
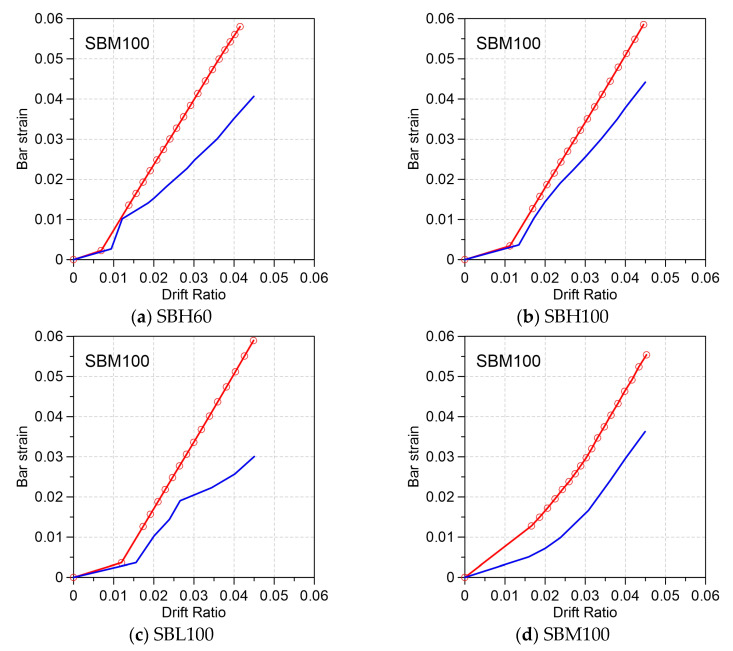
Modeled (red) and measured (blue) strain development of the longitudinal bar at the critical beam–joint interface: (**a**) Specimen SBH60; (**b**) Specimen SBH100; (**c**) Specimen SBL 100; (**d**) SBM 10.

**Figure 5 materials-16-04444-f005:**
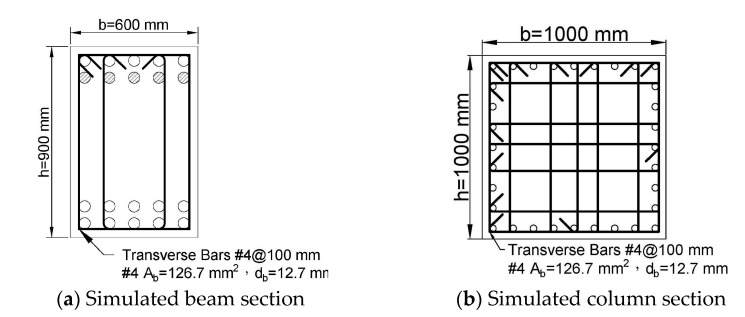
Simulated sections for the moment curvature analysis.

**Figure 6 materials-16-04444-f006:**
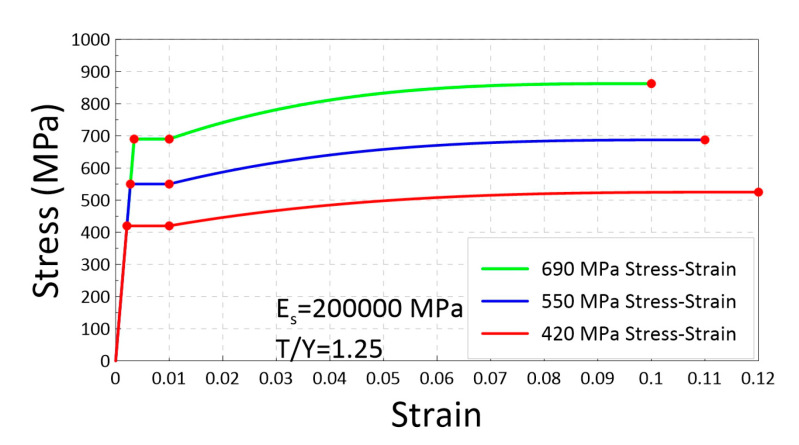
Idealized stress–strain curves for Grades 420, 550, and 690 for the moment–curvature analysis.

**Figure 7 materials-16-04444-f007:**
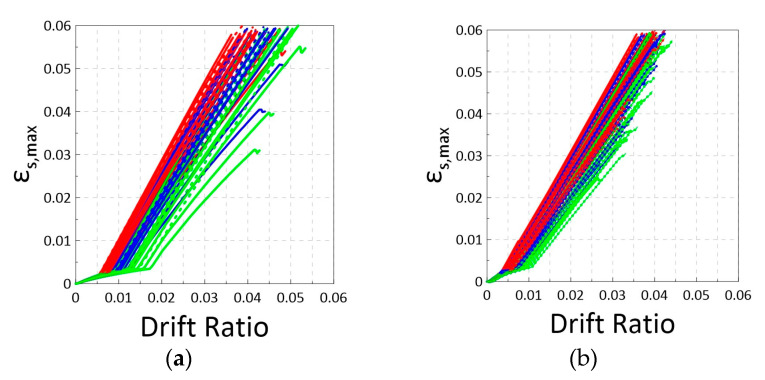
Relationships of the modeled bar strain at critical section versus drift ratio for (**a**) Simulated 72 beams with Grade 420 (red), 550 (blue), and 690 (green) bars. (**b**) Simulated 81 columns with Grade 420 (red), 550 (blue), and 690 (green) bars.

**Figure 8 materials-16-04444-f008:**
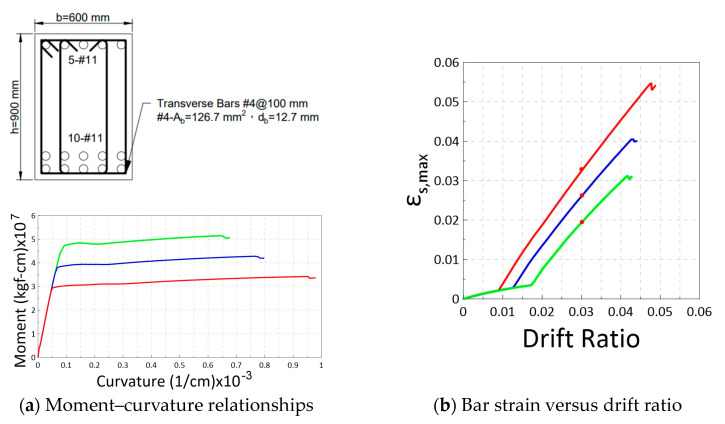
Effects of changing the bar $f_{y}$ (Grade 420 in red, 550 in blue, and 690 in green) for three beam sections with identical $f_{c}^{\prime }$ = 28 MPa and reinforcement ratios (10–#11(D36) tensile bars ρ= 2.0% and 5–#11(D36) compressive bars $\rho^{\prime }=$ 1.0%).

**Figure 9 materials-16-04444-f009:**
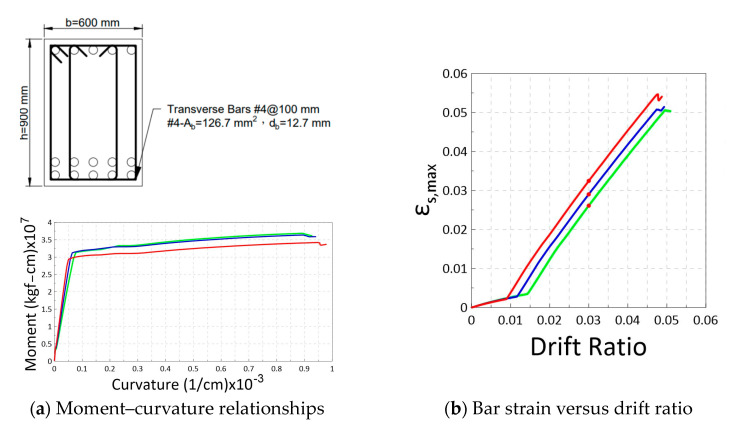
Effects of changing the bar $f_{y}$ and sizes for the three beam sections reinforced with Grade 420 (red), 550 (blue), and 690 (green) bars with similar moment strengths.

**Figure 10 materials-16-04444-f010:**
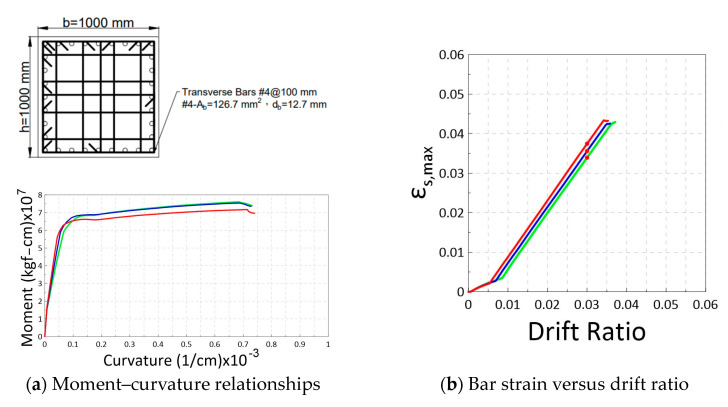
Effects of changing the bar $f_{y}$ and sizes for the three column sections reinforced with Grade 420 (red), 550 (blue), and 690 (green) bars with similar moment strengths.

**Figure 11 materials-16-04444-f011:**
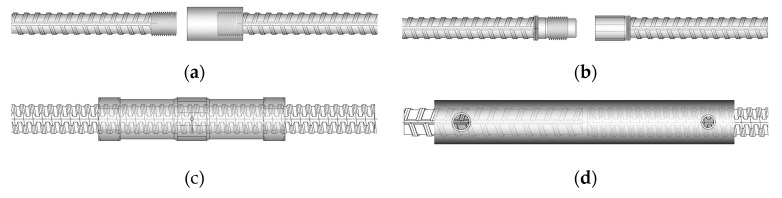
Four types of coupling systems evaluated: (**a**) threaded coupler with cold-forged, upsized bar threads; (**b**) friction-welded coupler; (**c**) grouted coupler for thread-deformed bars; (**d**) mortar-grouted coupling sleeve.

**Figure 12 materials-16-04444-f012:**
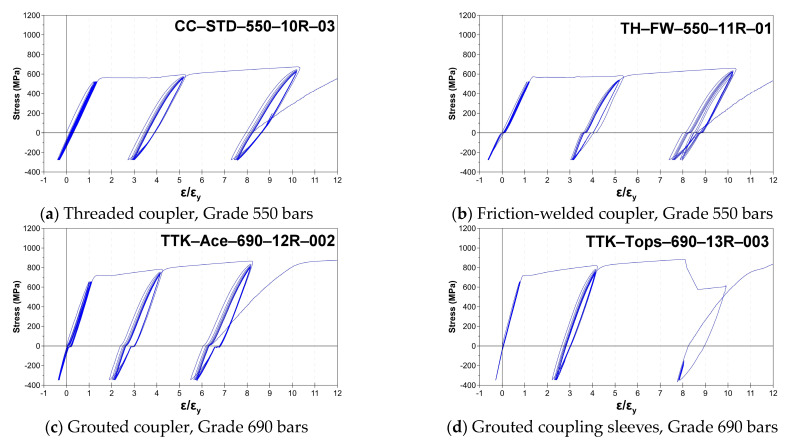
Typical elastic and inelastic cyclic responses of tested samples [38].

**Table 1 materials-16-04444-t001:** Categories of couplers in mechanical splices.

Applications ^1^	Category in ISO 15835-1	Category in JSCE	Category in Taiwan	Properties Tested
Basic	B	B	Type 1 (B)	Strength, ductility, and slip under static forces
Seismic 1—moderate	S1	A	Type 2 (A)	As for B + moderate low-cycle fatigue
Seismic 2— violent	S2	SA	Type 3 (SA)	As for B + violent low-cycle fatigue

^1^ Application for high-cycle fatigue. Category F of ISO 15835-1 is not shown.

**Table 2 materials-16-04444-t002:** Seismic loading protocols for simulating moderate and violent earthquakes.

Category	ISO15835-2:2009			JSCE 2007			Modification in Taiwan		
	Tension	Comp.	Cycles	Tension	Comp.	Cycles	Tension	Comp.	Cycles
S1 or A— elastic cyclic	0.90$f_{y}$	0.5$f_{y}$	20	0.95$f_{y}$	0.5$f_{y}$	20	0.95$f_{y}$	0.5$f_{y}$	16
S2 or SA— inelastic cyclic	2$\varepsilon_{y}$	0.5$f_{y}$	4	2$\varepsilon_{y}$	0.5$f_{y}$	4	$n\varepsilon_{y}$	0.5$f_{y}$	8
	5$\varepsilon_{y}$	0.5$f_{y}$	4	5$\varepsilon_{y}$	0.5$f_{y}$	4	$2n\varepsilon_{y}$	0.5$f_{y}$	8

**Table 3 materials-16-04444-t003:** Parameters of the simulated beam sections for the moment curvature analysis.

Parameter	Options			
	1	2	3	4
Concrete strength, $f_{c}^{\prime }$ (MPa)	28	49	70	-
Bar yield strength, $f_{y}$	420	550	690	-
Tensile reinforcement ratio, ρ	0.6%	1.0%	1.6%	2.0%
Tensile to comp. reinf. ratio, $\rho /\rho^{\prime }$	1.0	2.0	-	-

**Table 4 materials-16-04444-t004:** Parameters of the simulated column sections for the moment curvature analysis.

Parameters	Options		
	1	2	3
Concrete strength, $f_{c}^{\prime }$ (MPa)	28	49	70
Bar yield strength, $f_{y}$	420	550	690
Reinforcement ratio, ρ	1.0%	2.0%	3.0%
Axial load ratio, $P/A_{g}f_{c}^{\prime }$	0.10	0.20	0.30

**Table 5 materials-16-04444-t005:** Drift components at a 3% drift ratio for the beams shown in Figure 8.

Parameters ^1^ B_b ×$\;h\_f_{c\;}^{\prime }$$\_f_{y\;}$$\_\rho \;\_\;\rho^{\prime }$	$\Delta_{y}\;\mathrm{Components}$			$\Delta_{p}$	$\varepsilon_{s}$
	$\Delta_{y,v}$ (mm)	$\Delta_{y,s}$ (mm)	$\Delta_{y,f}$ (mm)	(mm)	at 3% Drift
B_60 × 90_28_420_2.0_1.0	1.6	7.7	30.6	95.1	0.0329
B_60 × 90_28_550_2.0_1.0	2.1	13.6	40.6	78.7	0.0262
B_60 × 90_28_690_2.0_1.0	2.6	22.0	52.9	57.5	0.0195

^1^ Character B represents beam section; **b**
× 
**h** in centimeters; $f_{c}^{\prime }$
and $f_{y}$
in MPa.

**Table 6 materials-16-04444-t006:** Drift components at a 3% drift ratio for the beams shown in Figure 10.

Parameters ^1^ B_b ×$\;h\_f_{c\;}^{\prime }$$\_f_{y\;}$$\_\rho \;\_\;\rho^{\prime }$	$\Delta_{y}\;\mathrm{Components}$			$\Delta_{p}$	$\varepsilon_{s}$
	$\Delta_{y,v}$ (mm)	$\Delta_{y,s}$ (mm)	$\Delta_{y,f}$ (mm)	(mm)	at 3% Drift
B_60 × 90_28_420_2.0_1.0	1.6	7.7	30.6	95.1	0.0329
B_60 × 90_28_550_1.6_0.8	1.8	11.6	38.7	82.9	0.0291
B_60 × 90_28_690_1.3_0.65	1.8	15.7	46.7	70.8	0.0258

^1^ Character B represents beam section; **b**
× 
**h** in centimeter; $f_{c}^{\prime }$
and $f_{y}$
in MPa.

**Table 7 materials-16-04444-t007:** Drift components at a 3% drift ratio for the columns shown in Figure 11.

Parameters ^1^ C_b ×$\;h\_f_{c\;}^{\prime }$$\_f_{y\;}$$\_\rho \;\_P/A_{g}f_{c}^{\prime }$	$\Delta_{y}\;\mathrm{Components}$			$\Delta_{p}$	$\varepsilon_{s}$
	$\Delta_{y,v}$ (mm)	$\Delta_{y,s}$ (mm)	$\Delta_{y,f}$ (mm)	(mm)	at 3% Drift
C_100 × 100_28_420_3.22_0.2	1.8	3.0	4.8	44.4	0.0374
C_100 × 100_28_420_2.61_0.2	1.9	4.5	6.1	41.5	0.0356
C_100 × 100_28_420_2.07_0.2	1.9	6.1	7.3	38.7	0.0339

^1^ Character C represents column section; **b**
× 
**h** in centimeter; $f_{c}^{\prime }$
and $f_{y}$
in MPa.

**Table 8 materials-16-04444-t008:** Tested couplers or coupling sleeves of Grade 550 and 690 reinforcements.

Grade	Mechanical Coupler or Sleeve			Samples	Category
	Producer	Type	Bar Size		
550	C.C.	Figure 11a	#8 (D25) #10 (D32)	3 3	SA SA
550	T.H.	Figure 11b	#11 (D36)	3	SA
550	B.L.C.	Figure 11b	#11 (D36)	3	SA
550	T.H.	Figure 11c	#11 (D36)	3	SA
550	T.T.K.	Figure 11c	#12 (D38) #13 (D41)	12 12	SA SA
550	T.T.K.	Figure 11d	#13 (D41)	12	A
690	T.H.	Figure 11c	#10 (D32)	3	SA
690	R.T.	Figure 11c	#10 (D32)	3	SA
690	T.T.K.	Figure 11c	#12 (D38) #13 (D41)	12 12	SA SA
690	R.T.	Figure 11d	#10 (D32) #10 (D32)	6 6	SA B
690	T.T.K.	Figure 11d	#13 (D41)	12	A

## Data Availability

The test data presented in this study are openly available in Reference [38].

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
