# Peer review of "Seismic Performance and Nonlinear Strain Analysis of Mechanical Splices for High-Strength Reinforcement in Concrete Structures"

_materials, 2023, doi:10.3390/ma16124444_

Round 1

Reviewer 1 Report (New Reviewer)

1.     In this form, the abstract is presented very concisely. Despite the structure, it will be difficult for the reader to immediately understand the objectives of the study, as well as the results obtained. It is necessary to slightly expand the amount of information presented in the annotation.

2.     The introduction given by the authors is too concise. Authors must submit a detailed literature review, from which a scientific problem should follow and, in accordance with it, the goal and objectives set. In the current version of the manuscript, the authors have not done this and must complete the introduction.

3.     The literature review should be supplemented with sources from 2022-2023. The topics considered by the authors are relevant and focused on practical application. There are a sufficient number of new literary sources that can be included in a literature review.

4.     It is necessary to fix the Figure 6 (a).

5.     The discussion of the obtained results is not presented in the form in which it is required in journals of this level. A detailed comparative analysis of the obtained results with the results obtained earlier by other authors should be carried out. The authors must give a detailed comparison, and only then the reader will understand the contribution of the authors to science and scientific novelty, and the new knowledge that has been obtained or the existing ideas that have been developed.

6.     In general, the article is interesting and scientifically developed, but the authors should strengthen the analytical component. The article is quite promising, but needs serious improvement. After completion, the article must be sent for re-reviewing.

Author Response

The authors would like to thank all the reviewers for their constructive comments, which provide the authors an opportunity to revise this article again and make it more readable. The reviewers’ comments and suggestions are consistent. The Abstract, Introduction, and Conclusions should be revised.  All the reviewers’ comments and suggestions are answered in the attached file. The corresponding changes made to revised manuscript are highlighted in red color and underline.  

Reviewer 2 Report (New Reviewer)

the study abstract should be shorten. study aim and objectives should be clearly stated after the introduction. more details about the experimental data must be highlighted after. new references related to study have to be added. also English language must be edited.

English language must be edited and checked.

Author Response

The authors would like to thank the reviewers for their constructive comments, which provide the authors an opportunity to revise this article and make it more readable. Here, all the reviewers’ comments and suggestions are answered in the attached file. The corresponding changes made to revised manuscript are highlighted in red text and underline.

Reviewer 3 Report (New Reviewer)

This study aimed at assessing the efficacy of mechanical and grouted splices in seismic concrete structures using nonlinear strain analysis and cyclic tests. The final objective of the authors is to modify the current acceptance criteria of mechanical splices for the use of high-strength reiforcement. The manuscript is interesting and offers new results and analysis. I have some minor comments suggestions for the authors before acceptance:

- The abstract does not provide the main findings of the study. Please include the main finding(s) of the study in the abstract. 

- Figure 2 is not needed and should be deleted.

- In the beginning of the study, the authors state that the main objective of the study is to modify the current acceptance criteria of mechanical splices for the use of high-strength reiforcement. Considering this objective, it seems that there is a lack of recommendations in the Conclusions to achieve such goal. Therefore, I invite the authors to add some recommendations in the conclusions to be used by designers.  

Author Response

The authors would like to thank the reviewers for their comments, which provide the authors an opportunity to revise this article and make it more readable. Here, all the reviewers’ comments and suggestions are answered in the attached file. The corresponding changes made to revised manuscript are highlighted in read texst and underline.

Round 2

Reviewer 1 Report (New Reviewer)

Thank you for your work on making corrections to the text of the article. All wishes were taken into account. The corrections made it possible to present the work you have done on the study at a decent level. Now the paper can be published. Good Luck!

Reviewer 2 Report (New Reviewer)

I would like to thank the authors for the amendments being made.

Reviewer 3 Report (New Reviewer)

All my comments have been addressed. The paper can be accepted for publication.

This manuscript is a resubmission of an earlier submission. The following is a list of the peer review reports and author responses from that submission.

Round 1

Reviewer 1 Report

A cyclic static load cannot be a seismic (dynamic) load model. This is a wrong concept. Under dynamic action, inelastic deformations of steel do not have time to fully manifest themselves. The material is brittle. This is especially true for high-strength reinforcement. Experimental dynamic studies are needed.

Author Response

The authors would like to thank the reviewers for their comments, which provide the authors an opportunity to revise this article and make it more readable. Here, all the reviewers’ comments and suggestions are answered. The corresponding changes made to revised manuscript are indicated and listed below.

~~~~~~~~~~~~~~~~~~~~~~~~~~~~~~~~~~~~~~~~~~~~~~~~~~~

(1) Response to the Reviewer #1.

C1: A cyclic static load cannot be a seismic (dynamic) load model. This is a wrong concept. Under dynamic action, inelastic deformations of steel do not have time to fully manifest themselves. The material is brittle. This is especially true for high-strength reinforcement. Experimental dynamic studies are needed.

A1: True. The behavior of reinforced concrete structures under earthquake excitations is unlikely simulated by quasi-static reversed cyclic loading tests. However, it is too difficult and pricy to carry out tests of large-scale structures on earthquake simulators (shaking table). Therefore, most of the large scale structural components are tested under quasi-static reversed cyclic loading.

To clarify the scope of this research is to modify the loading protocols for tensile test of mechanical splices, literature review of enhanced-ductility components of earthquake-resistant structures was added.

Please read Lines 97-143.

Reviewer 2 Report

This article presents in interesting way the nonlinear analysis for mechanical splices for high strength reinforcement in r/c structures.  The following changes/improvements are suggested.

- In the Abstract, please re-format the first sentence, because it starts abruptly.  

- Please check the text for small grammar and syntax issues.

- The conclusions section should be re-formatted to provide more mature conclusions, perhaps better numbered and mentioned separately to give more emphasis.

- In line 52, there is mentioned that the test programs and analysis are "elsewhere". Why have you split your work?  At least some important experimental points, results and conclusions, should be described in this work. 

- In line 108, the mentioned analysis program is not presented, while it is a not well-known/popular software.  Please describe it shortly, and give more references, and how it is realized. 

- Have you checked the role of the concrete quality in this work?  In other works, the quality of concrete is important for reinforced concrete elements. 

- Have you checked the proposed relationships/equations with other works?

- Similar works can be found in the literature, have you compared your work to other ones? Where is the novelty of this work?

Indicatively, these similar works are mentioned

1) Jianwei Zhang, Ruxing Cai, Chen Li, Xiao Liu, Seismic behavior of high-strength concrete columns reinforced with high-strength steel bars, Engineering Structures, Volume 218, 2020, 110861, ISSN 0141-0296, https://doi.org/10.1016/j.engstruct.2020.110861.

2) .M. Hassan, M. Elmorsy, Database trends and critical review of seismic performance tests on high strength steel reinforced concrete components, Engineering Structures, Volume 239, 2021,  112092, ISSN 0141-0296, https://doi.org/10.1016/j.engstruct.2021.112092."

Author Response

The authors would like to thank the reviewers for their comments, which provide the authors an opportunity to revise this article and make it more readable. Here, all the reviewers’ comments and suggestions are answered. The corresponding changes made to revised manuscript are indicated and listed below.

(2) Response to the Reviewer #2.

C0: This article presents in interesting way the nonlinear analysis for mechanical splices for high strength reinforcement in r/c structures.  The following changes/improvements are suggested.

C1: In the Abstract, please re-format the first sentence, because it starts abruptly.  

A1: Agreed. We completely rewrote the Abstract.

Please read Lines11-22. 

C2: Please check the text for small grammar and syntax issues.

A2: Yes. We corrected many grammar errors in the body text.

C3: The conclusions section should be re-formatted to provide more mature conclusions, perhaps better numbered and mentioned separately to give more emphasis.

A3: Yes. We corrected many grammar errors in the body text.

C4: In line 52, there is mentioned that the test programs and analysis are "elsewhere". Why have you split your work?  At least some important experimental points, results and conclusions, should be described in this work. 

A4: Sorry for misunderstanding. We did not split our work. Due to the length limitation, this paper presents the most important results obtained from the experimental program [8] and the numerical study [9]. The test setups and procedures of the test data presented in this paper are openly available in reference [8].

        Line 96 has been revised.

C5:  In line 108, the mentioned analysis program is not presented, while it is a not well-known/popular software.  Please describe it shortly, and give more references, and how it is realized. 

A5: Yes. Please read Lines 166-173. The revised text describes it shortly and cites popular software.

C6: Have you checked the role of the concrete quality in this work?  In other works, the quality of concrete is important for reinforced concrete elements. 

A6: Yes. The quality of concrete is important for reinforced concrete elements. In this study, cover (unconfined) and core (confined) concrete are modeled with the Mander’s model with some modifications (Lee et al. 2018). The measured concrete strengths were used in the modeling of beam tests.

C7: Have you checked the proposed relationships/equations with other works?

A7: Yes. Equations (1)-(5) is the plastic hinge model, which can be found in other works [19-22]. This study simply used the plastic hinge model to determine the lateral deformation from the moment-curvature relationships.

C8: Similar works can be found in the literature, have you compared your work to other ones? Where is the novelty of this work?

A8: Section 3.1, the plastic hinge model is similar to that in the literature. This section presents the basic theory and assumptions for modeling, which are not new at all.

Section 3.2, this study used the moment-curvature analysis program (NewRC-Mocur2020) and the plastic hinge model to predict the lateral load-deformation response of four beams tested by To and Moehle. This section demonstrates that the program and model are viable.

Section 4, Ductility Demands for Reinforcing Bars in Plastic Hinge Regions. The simulations presented herein is completely new with original contribution. Finally, the coupler tests demonstrated the modification on the loading protocols is valid.

C9: Indicatively, these similar works are mentioned

 1) Jianwei Zhang, Ruxing Cai, Chen Li, Xiao Liu, Seismic behavior of high-strength concrete columns reinforced with high-strength steel bars, Engineering Structures, Volume 218, 2020, 110861, ISSN 0141-0296, https://doi.org/10.1016/j.engstruct.2020.110861.

 2) .M. Hassan, M. Elmorsy, Database trends and critical review of seismic performance tests on high strength steel reinforced concrete components, Engineering Structures, Volume 239, 2021,  112092, ISSN 0141-0296, https://doi.org/10.1016/j.engstruct.2021.112092."

A9: Zhang et al. (2020) tests columns with high-strength steel bars. We could use the plastic hinge model to predict the envelops of lateral load-deformation curves. But, the measured strain readings reported by Zhang et al. (2020) is only available up to 0.005. The demand in plastic hinges of beams is relatively large. Therefore, we selected the beams tested by To and Moehle (2020) as benchmarks.

        Hassan and Elmorsy [10] reviewed prior cyclic tests of various high-strength rein-forced concrete components, including columns, beams, beam-column joints, walls, and coupling beams. A comprehensive database was established and evaluated. Has-san and Elmorsy [10] concluded that using high-strength steel instead of lower grade steel as longitudinal reinforcement based on matching flexural strength in beams and columns decreased hysteretic energy dissipation and drift capacity. Using high-strength steel as transverse reinforcement could achieve higher confining stress to enhance the strength and ductility in structural components.

        We cited the work of Hassan and Elmorsy [10] in Lines 102-105.

Reviewer 3 Report

1.      In the “Introduction” section, the authors shall explain the problem statement of this research, motivations, research gap, novelty, and implications of this study.

2.      The current study lacks the section “Literature review” to discuss the advantages of having ductile members in earthquake-resisting structures, also up-to-date practices to improve structural ductility should be addressed. The authors may consult (but are not limited to) the following references:

-        https://doi.org/10.14359/51700951

-        https://doi.org/10.1061/(ASCE)CC.1943-5614.0001218

-        https://doi.org/10.14359/51737144

3.      The methodology of this study is not clear, you may use a flow chart to describe the main steps you used to perform this study.

4.      It looks like you are using the designation of rebars sizes using the US style (Ex: #3, #4, #11…etc), you should either use only the international style (Ex: #25, #32, ….etc.) or use both of them together.

5.      In line 164, you said “plastic hinge model”, what is this model? You need to provide more details.

6.      For Figures 9a, 10a, and 11a, the authors should improve the quality of these figures.

7.      In line 166, you said “beam-joint interface”, what is this joint? You need to provide more details.

8.      Add a brief description of the database in figure 13.

9.      Add a new section to discuss the limitations and scope of this study.

10.   Add a new section to explain the feasibility of this work and its potential impact on practice.  

11.   For the section “Conclusions”, you should explain the problem statement, merits, motivations, and approach of this study.  The implications and feasibility of this work should also be discussed.

12.   The technical writing in this manuscript needs more improvements, the authors shall improve the writing of this manuscript considering the flow and coherence of discussions. 

Author Response

The authors would like to thank the reviewers for their comments, which provide the authors an opportunity to revise this article and make it more readable. Here, all the reviewers’ comments and suggestions are answered. The corresponding changes made to revised manuscript are indicated and listed below. 

(3) Response to the Reviewer #3.

C1: In the “Introduction” section, the authors shall explain the problem statement of this research, motivations, research gap, novelty, and implications of this study.

A1: Thanks for suggestion. We revised section 1. “Introduction” and added Section 2 “Review of enhanced-ductility components of earthquake-resistant structures.”

The problem statement of this research, research gap, novelty, and implications of this study was described.

Please read lines 27-50 and 136-143.

C2: The current study lacks the section “Literature review” to discuss the advantages of having ductile members in earthquake-resisting structures, also up-to-date practices to improve structural ductility should be addressed. The authors may consult (but are not limited to) the following references:

-        https://doi.org/10.14359/51700951

-        https://doi.org/10.1061/(ASCE)CC.1943-5614.0001218

-        https://doi.org/10.14359/51737144

A2: Thanks for suggestion. We added Section 2 “Review of enhanced-ductility components of earthquake-resistant structures” to discuss the advantages of having ductile members in earthquake-resisting structures, also up-to-date practices to improve structural ductility should be addressed.

Please read Lines 97-143.

C3: The methodology of this study is not clear, you may use a flow chart to describe the main steps you used to perform this study.

A3: Thanks for suggestion. Yes. Please read Lines 145-158.

C4:   It looks like you are using the designation of rebars sizes using the US style (Ex: #3, #4, #11…etc), you should either use only the international style (Ex: #25, #32, ….etc.) or use both of them together.

A4: Yes, thanks for suggestion. We revised the rebar size No. to show both of US and Metric styles. In the revised manuscript, they are #11(D36), #10(D32), #8(D25) etc.

C5: In line 164, you said “plastic hinge model”, what is this model? You need to provide more details.

A5: Yes, thanks for suggestion. We revised the descript of the plastic hinge model.

Please read Lines 160-164.

C6: For Figures 9a, 10a, and 11a, the authors should improve the quality of these figures.

A6: Yes, thanks for suggestion. We replaced them by higher-resolution pictures. Seems better.

C7:  In line 166, you said “beam-joint interface”, what is this joint? You need to provide more details.

A6: Yes, thanks for suggestion. Please read Line 233-234. We added a short description for the beam-joint interface.

C8:  Add a brief description of the database in figure 13.

A8: Yes, thanks for suggestion. Please read Lines 399 and 425. We added a new Table 8 to list the database.

C9:  Add a new section to discuss the limitations and scope of this study.

A9: Thanks for suggestion. The scope of this study was presented in Lines 136-143.

C10: Add a new section to explain the feasibility of this work and its potential impact on practice.  

A9: Thanks for suggestion. The feasibility of this work and its potential impact on practice has been presented in Section 5. “Modified Loading Protocols for Mechanical Bar Splices under Seismic Conditions”.

        Please read Lines 387-396.

    This study proposed an equal strain criterion, which has been adopted by the upcoming Design Code of Concrete Structures [6]. As shown in Table 2, the seismic loading protocol for Category SA mechanical splices has been modified to nεy and 2nεy, each for 8 cycles, with a strain ductility ratio of n=6, 5, and 4 for Grade 420, 550, and 690 bar splices, respectively. This modification would give an equal strain level of 2nεy=2.75% for the inelastic cycles of Grade 550 and 690 bar splices.

C11: For the section “Conclusions”, you should explain the problem statement, merits, motivations, and approach of this study.  The implications and feasibility of this work should also be discussed.

A11: Thanks for suggestion. The section of “Conclusions” was completely revised to address this comment.

C12: The technical writing in this manuscript needs more improvements, the authors shall improve the writing of this manuscript considering the flow and coherence of discussions. 

A11: Thanks for comment. Manuscript with major revision was submitted to the editorial office for consideration. The authors would like to revise this article to make it more readable and referable.

Round 2

Reviewer 1 Report

The authors write the following in their response: " it is too difficult and pricy to carry out tests of large-scale structures on earthquake simulators". This answer adds nothing to the scientific nature of the work. So, my comment still stands: “reject”.

Reviewer 2 Report

The authors have improved this manuscript, so it can be published.

Reviewer 3 Report

The authors have addressed the suggested comments.